# Down-Regulation of lncRNA MBNL1-AS1 Promotes Tumor Stem Cell-like Characteristics and Prostate Cancer Progression through miR-221-3p/CDKN1B/C-myc Axis

**DOI:** 10.3390/cancers14235783

**Published:** 2022-11-24

**Authors:** Ji Liu, Maskey Niraj, Hong Wang, Wentao Zhang, Ruiliang Wang, Aimaitiaji Kadier, Wei Li, Xudong Yao

**Affiliations:** 1Department of Urology, Shanghai Tenth People’s Hospital, School of Medicine, Tongji University, Shanghai 200072, China; 2Urologic Cancer Institute, School of Medicine, Tongji University, Shanghai 200331, China; 3School of Medicine, Tongji University, Shanghai 200331, China

**Keywords:** prostate cancer, cancer stem cells, MBNL1-AS1, CDKN1B, Wnt signaling pathway, single class logistic regression machine learning algorithm

## Abstract

**Simple Summary:**

To date, there has not been adequate research conducted on prostate cancer tumor stem cells, which play an important role in tumor recurrence, invasion, and drug resistance. In this study, we innovatively used a tumor stem cell-associated index (mRNAsi), calculated by a single-class logistic regression machine learning algorithm (OCRL), in combination with multiple databases to identify MBNL1-AS1 as a key lncRNA for prostate cancer stem cells and elucidated via in vivo and in vitro experiments. In addition, the experiments showed that the down-regulation of MBNL1-AS1 promotes the malignance of prostate cancer cell lines through the miR-221-3p/CDKN1B/C-myc axis by stimulating the stemness of tumor stem cells. This provides the basis for theoretical studies of prostate cancer and provides targets for precision therapy.

**Abstract:**

The recurrence, progression, and drug resistance of prostate cancer (PC) is closely related to the cancer stem cells (CSCs). Therefore, it is necessary to find the key regulators of prostate cancer stem cells (PCSCs). Here, we analyzed the results of a single-class logistic regression machine learning algorithm (OCLR) to identify the PCSC-associated lncRNA MBNL1-AS1. The effects of MBNL1-AS1 on the stemness of CSCs was assessed using qPCR, western blot and sphere-forming assays. The role of MBNL1-AS1 in mediating the proliferation and invasion of the PC cell lines was examined using Transwell, wounding-healing, CCK-8, EdU and animal assays. Dual-luciferase and ChIRP assays were used to examine the molecular mechanism of MBNL1-AS1 in PCSCs. MBNL1-AS1 was shown to be negatively correlated with stemness index (mRNAsi), and even prognosis, tumor progression, recurrence, and drug resistance in PC patients. The knockdown of MBNL1-AS1 significantly affected the stemness of the PC cells, and subsequently their invasive and proliferative abilities. Molecular mechanism studies suggested that MBNL1-AS1 regulates CDKN1B through competitive binding to miR-221-3p, which led to the inhibition of the Wnt signaling pathway to affect PCSCs. In conclusion, our study identified MBNL1-AS1 as a key regulator of PCSCs and examined its mechanism of action in the malignant progression of PC.

## 1. Introduction

Prostate cancer (PC) is the second most common malignancy in men, worldwide, and the fifth leading cause of cancer-related deaths, with approximately 1.4 million new cases and 400,000 deaths occurring worldwide each year [1]. Radical prostatectomy, brachytherapy, and external radiation therapy with follow-up monitoring are currently the most common treatments for localized PC [2]. For advanced or metastatic PC, androgen deprivation therapy (ADT) and chemotherapy are the main therapeutic approaches [3]. With standard treatment, 90% of patients achieve remission. However, a small percentage of patients will still develop castration-resistant prostate cancer (CRPC) or even metastatic castration-resistant prostate cancer (mCRPC) [4]. Therefore, there remains a critical need to examine the underlying molecular mechanisms of PC recurrence, progression, and drug resistance, and identify the corresponding therapeutic targets.

Multiple studies have reported the presence of a small subpopulation of cells called cancer stem cells (CSCs) in tumors, which are a source of tumor diversification and are essential for tumor development [5]. CSCs share unique molecular markers with normal stem cells, as well as functional similarities, such as the ability to self-renew and continuously generate heterogeneous tumor cells through asymmetric cell division [6,7,8]. The insensitivity of CSCs to radiotherapy, chemotherapy and targeted drugs means that even though the majority of tumor cells are eliminated by these treatments, the remaining CSCs can restart tumor growth and invasion through proliferation and differentiation [9]. Thus, CSCs have been closely associated with tumor progression, metastasis, recurrence, and drug resistance, stemming from the insensitivity of CSCs to radiotherapy, chemotherapy and even targeted drugs [5,10]. PC is a heterogeneous disease and contains many cancer cell subtypes, including prostate cancer stem cells (PCSCs) [11].

Recent studies have focused on the characterization of cancer stemness using artificial intelligence and deep learning methods. For example, using a single-class logistic regression machine learning algorithm (OCLR), Tathiane et al. [12] analyzed the molecular characteristics of cell types with different degrees of stemness. The stemness index was derived from the OCLR algorithm, trained on stem cell classes (ESC/iPSC; SC), and differentiated endodermal, mesodermal, and ectodermal progenitors. In addition, stemness was quantified by combining multi-platform analyses of the transcriptome, transcription factor binding sites, and methylome, and ultimately identifying a stemness index based on mRNA expression (mRNAsi) and DNA methylation (mDNAsi). The mRNAsi scores have been positively correlated with the tumor dedifferentiating properties of CSCs. Here, we have applied this method to the PC dataset of The Cancer Genome Atlas (TCGA) to derive the mRNAsi and mDNAsi scores for the PC samples.

Long non-coding RNAs (lncRNAs) are essentially RNA transcripts ≥ 200 bp that do not encode proteins. The number of non-coding genes identified in recent years now exceeds the number of coding genes. However, the function of most lncRNAs is unknown [13,14]. Known lncRNAs have been shown to influence gene expression by acting as microRNA (miRNA) sponges and scaffolds for molecular complexes, activators or decoys of transcription factors, and recruiters of chromatin-modifying complexes. Therefore, lncRNAs have a range of regulatory roles in the functional activities of cancer cells, including tumorigenesis, development and even CSCs [15].

Although there are abundant and diverse treatment options for PC, there is still a lack of effective methods to prevent the recurrence and progression of PC. This study aimed to analyze an extensive database of samples using an OCLR machine-learning algorithm to identify lncRNAs closely associated with the formation of PCSCs and to elucidate the potential molecular mechanisms by which PCSCs promote tumor progression, in order to identity corresponding therapeutic targets.

## 2. Materials and Methods

### 2.1. Data Processing Software

Our study is outlined in the flow chart shown in Appendix A. R version 3.6.2 (Action of the Toes) with Strawberry Perl version-5.14.2.1 (64-bit) (URL: https://www.perl.org/) (accessed on 9 January 2020) was used in this study. All R packages used were sourced from public websites and downloaded from Bioconductor (URL: http://www.bioconductor.org/) (accessed on 9 January 2020). All statistical and data processing software is open source and can be downloaded at the URL provided above. Prism software version (http://www.graphpad-prism.cn) (accessed on 9 January 2020) was used to present the results of the data analysis.

### 2.2. Database and mRNAsi Index

The transcriptomic RNA sequencing datasets [expressed as fragments per kilobase of exons per million reads mapped (FPKM)] of 551 patients with PC (499 cases of tumor tissue and 52 cases of normal tissue) were downloaded from TCGA database (https://cancergenome.nih.gov/) (accessed on 9 January 2020). The clinical data of these PC patients were derived from UCSC (http://genome.ucsc.edu/index.html) (accessed on 9 January 2020), and cases that lacked complete data were not included. The mRNAsi index of TCGA-derived PC samples described by Tathiane et al. [12] was merged with the TCGA PC clinical data using a Strawberry Perl script, and cases that did not match with normal tissue were removed. The Wilcoxon test was used to analyze differences in mRNAsi between PC subtypes. Subcellular localization of lncRNAs was analyzed using the lncATLAS database (http://lncatlas.crg.eu/) (accessed on 9 January 2020) to predict potential regulatory mechanisms of lncRNAs. The RAID database (http://www.rna-society.org/raid/index.html) (accessed on 9 January 2020) was used to predict lncRNAs that act as miRNAs. Comparative analysis of the predicted lncRNAs with differentially expressed miRNAs in the TCGA database was used to identify candidate target genes. Using RAID, miRDB (http://www.mirdb.org/) (accessed on 9 January 2020), mirDIP (http://ophid.utoronto.ca/mirDIP/) (accessed on 9 January 2020), TargetScan (http://www.targetscan.org) (accessed on 9 January 2020), and miRTarBase (http://mirtarbase.mbc.nctu.edu) (accessed on 9 January 2020) combined with co-expression analysis data from the TCGA-PRAD dataset, we screened for protein-coding RNAs downstream of the miRNAs. Our results were presented using jvenn (http://jvenn.toulouse.inra.fr/app/ example.html) (accessed on 9 January 2020).

### 2.3. Screening of Differentially Expressed mRNAs

Differentially expressed mRNAs between PC and normal samples in the TCGA database were analyzed using edgeR software (http://bioconductor.org/package/edgeR/) (accessed on 9 January 2020). The raw data were normalized using the edgeR Bioconductor package, and differences between the two sets of gene expression data were analyzed with the cut-off data of log2 |fold change| > 1 and false discovery rate (FDR) < 0.01.

### 2.4. Weighted Gene Co-Expression Network Approach (WGCNA)

Using the “WGCNA” R package v1.69, gene co-expression networks were constructed based on the expression matrix of mRNAs. First, the Pearson’s correlation coefficient matrix of all paired genes was constructed using gene co-expression similarity. When the degree of independence (R2) was 0.9, the appropriate β value was determined to generate a scale-free network. Subsequently, the weighted adjacency matrix was transformed into a topological overlap matrix (TOM), which measures network connectivity. We used a gene dendrogram with a minimum module size of 50, and highly similar modules with a correlation higher than 0.7 were merged. Finally, genes with similar expression profiles were grouped into gene modules by a mean linkage hierarchical clustering method based on differences in TOM measurements. All genes were represented by the expression of module eigengene (ME) in a specific module. The module most highly associated with mRNAsi (|r| = 0.82) was selected for further analysis. Gene Ontology (GO) and Kyoto Encyclopedia of Genes and Genomes (KEGG) analyses were carried out to identify the potential molecular pathways and physiological processes associated with the screened mRNAsi-related genes.

### 2.5. Cell Culture

Normal human prostate epithelial (RWPE-1) and CRPC [Castration-Resistant Prostate Cancer (PC3, DU-145)] cell lines were obtained from the Chinese Academy of Sciences. The CRPC cell lines were cultured in RPMI-1640 (Gibco, New York, NY, USA) containing 10% fetal bovine serum and 1% penicillin/streptomycin. The RWPE-1 cell line was cultured in Defined Kerationocyte-SFM (Invitrogen, Waltham, MA, USA) containing 1% keratinocyte growth supplement. All cells were cultured at 37 °C and 5% CO_2_.

### 2.6. Real Time Quantitative PCR (RT-qPCR)

The total RNA was extracted from the cells using TRIzol^®^ reagent (Thermo Fisher Scientific Inc., Waltham, MA, USA) according to the manufacturer’s instructions. RT-qPCR was performed using SYBR^®^Green PCR Master Mix (Thermo Fisher Scientific Inc., Waltham, MA, USA) and a 7900HT Fast Real-Time qPCR System with the following cycling conditions: 95 °C for 10 min, 95 °C for 10 min, followed by 40 cycles at 95 °C for 10 s and 60 °C for 1 min. The relative mRNA levels were normalized using GAPDH as a control. The 2−ΔΔCt method was used to analyze the results. The sequence of the primer is shown in Appendix A.

### 2.7. Enrichment of CSCs

The PC3 and DU-145 cells were cultured in serum-free Dulbecco’s Modified Eagle culture (DMEM)/F12 culture (Gibco, New York, NY, USA) supplemented with insulin (Sigma-Aldrich, St. Louis, MO, USA), 20 ng/mL human recombinant epidermal growth factor (Peprotech, Suzhou, China), 10 ng/mL alkaline fibroblast growth factor (Peprotech, Suzhou, China) and 0.4% bovine serum albumin (BSA). After 2 weeks, 3 × 10^5^ cells were seeded into six plates to assess the enrichment of CSCs. The medium was changed every other day, and changes in the morphology of the CSCs were visualized by microscopy.

### 2.8. Fluorescence in Situ Hybridization (FISH)

The subcellular localization of MBNL1-AS1 was detected using a FISH kit (BIS-P0001, Guangzhou Boxin Biotechnology Co., Ltd., Guangzhou, China) according to the manufacturer’s instructions. Images of the samples were acquired under a Zeiss LSM880 NLO (2 + 1 with BIG) confocal microscope (Leica Microsystems, Wetzlar, Germany). The experiment was performed three times. The sequence of the probe is shown in Appendix A.

### 2.9. Subcellular Fractionation

Chromatin fractionation was performed using the PARIS kit (Life Technologies, Carlsbad, CA, USA) according to the manufacture’s instructions.

### 2.10. Western Blotting Assay

Cellular proteins were extracted using the radio-immunoprecipitation assay (RIPA) (Thermo Fisher, Waltham, MA, USA), then separated by 10% SDS-PAGE, and transferred to 0.22 μm PVDF membranes (Millipore, Burlington, MA, USA). Membranes were blocked with 5% skimmed milk solution and incubated with specific primary antibodies overnight at 4 °C. After washing, membranes were then incubated with specific secondary antibodies for 1 h. The silver stain detection system (Beyotime, Nantong, China) was used to detect the protein bands. GAPDH was used as the internal control. The detail of antibody involved in this experiment is showed in Appendix A. And all of the original uncropped western blot was shown in the Appendix A.

### 2.11. Transfection of Cell

The MBNL1-AS1 three-target interference plasmid was synthesized by Genomeditech (Nantong, China). Lentiviral vector stable transfection strains were constructed using the ZR-LPK-002 kit (ZORIN, Shanghai, China) with 293T cells according to the manufacturer’s instructions. Collected lentiviral particles were transfected into PC cell lines after screening for stably transfected cell lines using puromycin (5 mg/mL). The siRNA and miR-mimics were obtained from Genomeditech (Shanghai, China). Transfection was performed using Lipofectamine 3000 (Thermo Fisher, Waltham, MA, USA).

### 2.12. Sphere-Forming Assay

PC3 and DU145 cells were seeded into six-well ultra-low cluster plates (Corning, NY, USA) according to the number of 1000 cells per well, respectively. Then, these cells were cultured with DMEM/F12 serum-free medium (Invitrogen) supplemented with 20 ng/mL bFGF (Beyotime, Nantong, China), 2% B27 (Thermo Fisher, Waltham, MA, USA), 20 ng/mL EGF (Beyotime, Nantong, China), 5 μg/mL insulin (Beyotime, Nantong, China) and 0.4% BSA (Sangon, Shanghai, China). A count and photograph of spheres were taken after two weeks. (Cell clusters larger than 50 mm in diameter were considered to be CSC)

### 2.13. Wounding Healing Assay

The PC cells were seeded into 6-well plates at a density of 5.0 × 10^4^ cells/cm^2^. When the cells reached 80% confluency, scratches (10 μm) were made across the cellular layer. The plates were washed twice with phosphate-buffered saline (PBS), then the cells were incubated in RPMI-1640 medium containing 3% FBS in a saturated atmosphere of 5% CO_2_ at 37 °C for 24 h. Images at the 0 h and 24 h time points were obtained using an inverted microscope. ImageJ software was used to calculate the 24 h migration distance (cell migration distance was calculated by subtracting the distance between the scratch edge at 0 h and the migration edge at 24 h). Each set of experiments was repeated three times.

### 2.14. Transwell Assay

Polyester membrane cell embedding dishes with 24-well plates (insert: 8.0 μm; diameter: 6.5 mm, JET, Guangzhou, China) were used to examine the invasive and migratory abilities of PC cells. Next, 6.0 × 10^4^ cells were resuspended in serum-free DMEM and seeded into the upper chamber, which had been pre-coated with 200 mg/mL Matrigel (1:8, Yeasen, Shanghai, China). DMEM containing 10% FBS was placed in the lower chamber. After 16 h incubation, cells that had invaded the lower chamber through the Matrigel were fixed with 90% ethanol for 10 min and stained with 0.5% Crystal Violet for 15 min. Five randomly selected fields of view were observed and the number of invaded cells were counted under the microscope. This experiment was repeated three times.

### 2.15. EdU Incorporation Assay

The stably transfected cells were seeded into 96-well plates at a density of 1 × 10^4^ cells/well. An EdU binding assay kit (Beyotime, Nantong, China) was used to assess cell proliferation. Images of stained cells were captured using a fluorescent microscope (Nikon, Tokyo, Japan). The experiment was repeated three times.

### 2.16. Cell Counting Kit-8 (CCK-8) Assay

The cells were seeded into 96-well plates at a density of 1.0 × 10^3^ cells/well and incubated at 37 °C in an atmosphere of 5% CO. At various time points (24, 48, 72 and 96 h), 10 μL CCK-8 reagent was added, and the optical density (OD) values were measured at a wavelength of 450 nm. This experiment was repeated three times.

### 2.17. Dual-Luciferase Assay

The luciferase plasmids of wild-type or mutant MBNL1-AS1, miR-221-3p and CDKN1B (Aibosi, Shanghai, China) were co-transfected with miR-mimics in HEK293T cells using Lipo3000. A dual-luciferase reporter gene kit (Yeasen, Shanghai, China) was used to measure luciferase activity. This experiment was repeated three times. The experiment design of the pmirGlo vector is shown in Appendix A.

### 2.18. ChIRP Assay

A Chromatin Isolation by RNA Purification (ChIRP) Kit (Guangzhou Boxin Biotechnology Co., Ltd., Guangzhou, China) was used to detect MBNL1-AS1 and miR-221-3p, as well as miR-221-3p and CDKN1B binding. Complementary probes for biotin-labeled miR-221-3p were co-incubated with cell lysates according to the manufacturer’s instructions. Finally, the bound mRNAs were analyzed by RT-qPCR.

### 2.19. Animal Experiments

All animal experiments were authorized by the Ethics Committee of the Tenth People’s Hospital, Tongji University, and were in accordance with the National Institutes of Health Guidelines for the Care and Use of Animals and the principles of the Declaration of Helsinki. Approximately 8.0 × 10^6^ cells were mixed into 200 µL Matrigel (1:8, Yepsen, China) and injected subcutaneously into 8 male BALB/c nude mice (age 5–6 weeks, weight 20–22 g, Experimental Animal Operations Department, Shanghai Institute of Family Planning Science, Shanghai, China) without specific pathogens. At week 4 after the experiment, the nude mice were euthanized, tumors were excised and photographed, and preserved in 4% paraformaldehyde for subsequent immunohistochemical experiments.

### 2.20. Nomogram Development and Validation

Univariate and multivariate Cox regression analyses were used to construct nomograms to predict the impact of target genes and other clinicopathological features on prognosis and metastasis in PC patients. The nomogram was validated by calibration plots and consistency index (C-index) using the rms package v5.1 in R (https://CRAN.R-project.org/web/packages/rms/index.html) (accessed on 9 January 2020).

### 2.21. Data Analysis

Heat maps and volcano maps were drawn using the heatmap package (https://cran.r-project.org/web/packages/pheatmap/index.html) (accessed on 9 January 2020). Disease progression was used as an endpoint with clinical data obtained from the TCGA database. Survival analysis was performed in log2 ^(normalized value + 1)^ data format. The effect of genes on progression-free survival (PFS) was identified by univariate Cox analysis using the R survival package. The sensitivity and specificity of ROC curves were used to assess the reliability of genes as prognostic factors. The area under the survival ROC curve for risk scores was calculated using the survival ROC software package. Independent *t*-tests were used to test for differences between clinical parameters. A *p*-value less than 0.05 was considered to be statistically significant.

## 3. Results

### 3.1. Identification of CSC-Associated mRNAs

The flow chart of identifying CSC-associated lncRNA is shown in Appendix A. We identified differentially expressed genes between PC and healthy samples by differential analysis (*p*-value < 0.05, log2 |fold change| > 1). The heat map and volcano plot, showing 2521 differentially expressed mRNAs in PC samples versus normal samples, are shown in Appendix A. Box plots displaying significant differences in the mRNAsi scores between the PC samples and normal tissues are shown in Figure 1A. CSC-associated lncRNAs were identified using the WGCNA method to exclude samples that showed large differences by clustering analysis. A total of 452 samples were identified for subsequent analysis. During the downscaling analysis, β value = 7 (free ratio R2 = 0.9) was considered to be the optimal soft threshold. Six gene modules were subsequently identified by gene clustering methods (Figure 1C), with the brown module (r = −0.82, *p*-value < 0.001) having the strongest correlation with mRNAsi (Figure 1D). Due to the close correlation of the genes in this module with the module itself (r = 0.97; *p*-value < 0.001) (Figure 1E), the brown module was selected as a target module. Finally, 1221 mRNAsi-related mRNAs were extracted from the module, and the final CSC-associated lncRNAs were obtained by gene classification.

### 3.2. Identification and Validation of mRNAsi-Related lncRNAs

The PC3-CSC cells were isolated and enriched by culturing the PC3 cells in serum-free culture conditions. Microscopic examination revealed that the isolated cells grew in a spherical distribution, characteristic of CSCs (Figure 2A). Next, western blot analysis and qPCR were performed to examine changes in the expression levels of relevant stemness markers including CD133, KLF4, NANOG, and OCT4 (Figure 2B,C). The top five lncRNAs most strongly associated with mRNAsi in the normal cell line RWPE-1 compared to the PC3 and DU-145 tumor cell lines are shown in Figure 2D–F, with the MBNL1-AS1 expression consistent with the predicted results. Subsequent analysis revealed that MBNL1-AS1 was expressed at low levels in the DU-145 and PC3 cell lines, while significantly lower expression was observed in the CSCs (Figure 2G,H). The survival curves for the TCGA-PRAD cohort (samples with missing clinical characteristics were removed) showed significant correlation between MBNL1-AS1 and the progression free interval (PFI) of the PC patients (Figure 2I) (*p*-value < 0.05). Correlation analysis for target genes and clinical characteristics showed a significant negative correlation between MBNL1-AS1 and local tumor invasion, lymphatic metastasis, recurrence and drug resistance, with significant clinical implications (Figure 2J).

### 3.3. Localization of Target Genes and Their Effects on Stem Cells

Analysis of multiple GEO (GSE68555, GSE32571 and GSE29079) and TCGA datasets further validated the differential expression of MBNL1-AS1 in cancer and normal tissues (Appendix A). According to the lncATLAS website, MBNL1-AS1 was predominantly localized in the cytoplasm, suggesting that its target genes would also be found in the cytoplasm (Appendix A). Subsequent FISH and nucleoplasmic separation assays confirmed this prediction; then, we silenced the MBNL1-AS1 expression in the PC3 and DU-145 cell lines using three lentiviral plasmids. The knockdown efficiency of sh1-MBNL1-AS1 and sh2-MBNL1-AS1 were found to be the most effective (Figure 3A–F). Next, we sought to determine the role of MBNL1-AS1 and its target genes on CSC function. The knockdown of MBNL1-AS1 was found to significantly enhance the sphere-forming ability of the CSCs (Figure 3G). Furthermore, MBNL1-AS1 knockdown significantly enhanced the stemness of the CSCs, as demonstrated by the changes in the expression levels of stemness-related proteins and genes (Figure 3H,I).

### 3.4. Knockdown of MBNL1-AS1 Increases the Proliferative and Invasive Abilities of PC Cells

Next, the effects of silencing the MBNL1-AS1 expression on PC proliferation and invasion were examined. Using CCK-8 assays, we demonstrated that knockdown of MBNL1-AS1 significantly enhanced the cell viability of the PC3 and DU-145 cells (Figure 4A,B). Furthermore, flow cytometric analysis of the cell cycle revealed that the knockdown of MBNL1-AS1 resulted in significant S phase arrest (*p*-value < 0.05) (Figure 4C,D) In addition, EdU, Transwell, and scratch assays revealed that silencing the MBNL1-AS1 expression was associated with a significant increase in the proliferative and invasive abilities of the PC cells, both in vitro in PC cell lines and in vivo in a subcutaneous tumorigenesis mouse model (Figure 4F–K). Our immunohistochemical results further supported these findings (Figure 4L,M). Finally, we found that the protein expression of the proliferation markers PNCA and Ki-67, in addition to the invasion-associated markers Snail and N-Cadherin, were significantly increased in the MBNL1-AS1-silenced cells. Thus, taken together, our findings indicated that the knockdown of MBNL1-AS1 led to increased proliferation and invasiveness of the PC cells (Figure 4N).

### 3.5. MBNL1-AS1 May Affect PC and CSCs through miR-221b-3p-Targeted CDKN1B

Based on its subcellular localization, we hypothesized that MBNL1-AS influences downstream target genes in PC and CSCs through the competitive adsorption of miRNAs. Thus, we combined the RAID and TCGA-PRAD databases to predict miRNAs regulated by MBNL1-AS1. As shown in the Venn diagram, three miRNAs, including hsa-miR-221-3p, hsa-miR-218-5p, and hsa-miR-30a-5p, were predicted (Figure 5A). Next, we examined the effects of MBNL1-AS1 knockdown on the expression levels of these miRNAs and found that miR-221-3p expression was significantly upregulated (Figure 5B). Then, we constructed a dual luciferase plasmid of MBNL1-AS1 (Figure 5C). Dual-luciferase and pull-down experiments confirmed that MBNL1-AS1 and miR-221-3p interacted (Figure 5D,E). Thus, our findings suggested that MBNL1-AS1 may act as a molecular sponge, competitively adsorbing miR-221-3p, and potentially contributing to PC malignancy. Next, the mirDIP, miRDB, RAID, TargetScan, miRTarBase, and TCGA databases were combined to predict the downstream target genes of miR-221-3p. Venn diagram analysis suggested that CDKN1B was the only gene that met the screening criteria. Thus, CDKN1B was selected for further studies (Figure 5F). Using qPCR, we demonstrated that decreased miR-221-3p expression was associated with increased CDKN1B expression (Figure 5G). Then, we constructed a dual luciferase plasmid of CDKN1B (Figure 5H). Using dual-luciferase and ChIRP assays, we confirmed a binding relationship between miR-221-3p and CDKN1B (Figure 5I,J). These findings demonstrated that MBNL1-AS1 could competitively bind to miR-221-3p, which then targeted CDKN1B, potentially contributing to the malignancy of the PC cells and PCSCs. We further validated the effects of the MBNL1-AS1 downstream target genes on the PC cell lines by using EdU and Transwell assays to demonstrate that miR-221-3p attenuated the invasive and proliferative abilities of the PC3 and DU-145 cells (Figure 5K–M). Furthermore, using western blot analysis, we showed that altering the transcript levels of MBNL1-AS1 and miR-221-3P led to changes in the CDKN1B expression levels (Figure 5N). Finally, we found that the CDKN1B was negatively correlated with FPS and mRNAsi in prostate cancer patients, and subsequent stem cell sphere-forming experiments validated these results (Figure 6A–C).

### 3.6. MBNL1-AS1 Regulates the Wnt Pathway through the miR-221-3p/CDKN1B/C-m yc Axis in PCSCs

To further determine the underlying mechanism of MBNL1-AS1 in PC, we divided 499 tumor samples in the TCGA-PRAD dataset into high and low expression groups, based on the transcriptomic levels of MBNL1-AS1. A total of 521 differential genes were obtained by differential analysis (*p*-value < 0.05, log2 |fold change| > 1). GO and KEGG analyses identified “Muscle transfer” as the most relevant function and pathway (Figure 6D). GSEA analysis indicated that the Wnt signaling pathway was linked to both MBNL1-AS1 and CDKN1B (Figure 6E,F). Correlation analysis of WNT3A with C-myc revealed that C-myc was significantly and positively correlated with CDKN1B in the TCGA-PRAD cohort (R = −0.15, *p* <0.01) (Figure 6G,H). Subsequent western blot analysis confirmed the correlation between these two genes (Figure 6I). In conclusion, our findings demonstrated that MBNL1-AS1 mediated the Wnt signaling pathway through the miR-221-3p/CDKN1B/C-myc axis (Figure 7).

### 3.7. Validation of the Effect of MBNL1-AS1 and CDKN1B on Drug Sensitivity in Prostate Cancer

The prediction results for MBNL1-AS1 sensitivity (50% inhibitory concentration, IC50) suggested that the MBNL1-AS1 high expression group was more sensitive to Ipatasertib drugs and common chemotherapy drugs for prostate cancer, such as Gemcitabine and Docetaxel, whereas the MBNL1-AS1 low expression group was sensitive to Wnt signaling pathway inhibitor drugs, such as Paclitaxel_1080, LGK974_1598 (Figure 8A,B). Similarly, the prediction results for CDKN1B sensitivity (50% inhibitory concentration, IC50) suggested that the CDKN1B high expression group was more sensitive to JQ1_2172 drugs and common chemotherapy drugs for prostate cancer, such as Gemcitabine and Docetaxel. In contrast, the CDKN1B low expression group was sensitive to Wnt signaling pathway inhibitor drugs, such as Paclitaxel_1080, LGK974_1598 and Wnt-C59_1622 (Figure 8C,D).

### 3.8. Establishment and Validation of Nomograms to Determine the Clinical Value of MBNL1-AS1 in the Prognosis and Potential Metastasis of PC

To further evaluate the clinical value of MBNL1-AS1 in the analysis of the PRAD samples in the TCGA-PRAD cohort, clinical characteristics were screened by univariate and multivariate Cox regression. In addition, the PFS at 1, 3 and 5 years was predicted based on the MBNL1-AS1 gene expression levels, Gleason score and T-stage, created in a nomogram (Figure 9A). The actual C-index obtained was 0.81 ± 0.17 (mean ± standard error), indicating that the actual PFS of the sample was generally consistent with the predicted PFI (Figure 9B). Finally, the calibration curves suggested that the nomogram was a good predictor of PFI (Figure 9C–E). Similarly, for distant metastasis in the PRAD samples, we generated a nomogram to predict tumor invasion and progression. After screening by univariate and multifactorial Cox regression analyses, we included the gene expression level of MBNL1-AS1, pathological T-stage, and N-stage into the model (Figure 9F,G). Our results revealed a nomogram C-index of 0.835 ± 0.05 (mean ± standard error), indicating that the actual distant metastasis rate was generally consistent with the predicted results. Calibration plots also showed agreement between the nomogram predictions and true distant metastases (Figure 9H).

### 3.9. MBNL1-AS1 Enhances Mitochondrial Activity in PC Cells through the Synergistic Action of SEPT4

Next, we examined the relationship between MBNL1-AS1 and mitochondrial activity using reactive oxygen species (ROS) assays. We found that sh-MBNL1-AS1 treatment led to elevated ROS levels compared to the normal control (NC) group (Appendix A), suggesting that MBNL1-AS1 may inhibit mitochondrial function in tumor cells. To determine the molecular mechanisms involved, we downloaded mitochondria-related genes (n = 1200) from the Turbot website and divided these genes into 16 modules by WGCNA analysis (Appendix A). Correlation analysis revealed that the genes in the black module were the most significantly correlated with MBNL1-AS1 (R = 0.72 Cor < 0.001) (Appendix A), with high module confidence (Cor = 0.79, *p*-value < 0.001) (Appendix A) (HEBP1 also is a core gene of mitochondrial activity). Comparative analysis indicated that SEPT4 (in the black module gene list) was associated with cell stemness. Correlation analysis suggested that SEPT4 was significantly positively correlated with MBNL1-AS1 (R = 0.37, *p*-value < 0.001) (Appendix A). These findings were confirmed by western blot analysis (Appendix A).

### 3.10. SEPT4 Is Closely Associated with PC Progression and Recurrence

To determine the correlation between SEPT4 and the malignant progression of PC, we carried out correlation analysis using the SEPT4 expression data from the TGCA-PRAD dataset and the pathological T-stage and N-stage of PC. We found that SEPT4 was closely associated with recurrence, lymph node metastasis, and the Gleason pathological grade of PC (Appendix A). Next, to identify the potential biological functions and pathways associated with SEPT4, we performed GO and KEGG analysis using GSEA. We found that SEPT4 may be involved in regulating the cAMP metabolic and circulating nucleotide metabolic processes (Appendix A), as well as the Wnt and TGF-β signaling pathways (Appendix A).

### 3.11. MBNL1-AS1 Is Significantly Associated with the Immune System and Immunotherapy Responsiveness

The TCGA-PRAD dataset was used to determine the relationship between MBNL1-AS1 and the immune system. The distribution of the tumor infiltrating immune cells (TIICs) in 499 PC samples is shown in Appendix A. Box plots demonstrated significant differences between the low- and high-expression MBNL1-AS1 groups in the activation of tumor-associated immune responses, such as the activation of immune checkpoints, co-stimulation of antigen-presenting cells, cytolytic activity, type I and type 2 interferon responses, inflammation promotion, T-cell co-stimulation and suppression, and the activation of human leukocyte antigens (*p*-value < 0.001) (Appendix A). In the TCGA-PRAD cohort, the effect of MBNL1-AS1 on immunotherapy response and immune efflux was verified by applying the TIDE algorithm. We found that immune nonresponse and immune escape were significantly higher in the low-expression MBNL1-AS1 group than in the high expression group (χ2 test, *p*-value < 0.001; Appendix A). Box plots were used to show differences in the level of immune cell infiltration in the low- and high-expression scoring subgroups. We found that the levels of activated CD8+ T cells, CD4+ T cells, macrophages, dendritic cells, mast cells, tumor-infiltrating lymphocytes (TIL), T follicular helper cells, helper T cells, and Tregs were significantly different between the two groups (*p*-value < 0.05) (Appendix A).

## 4. Discussion

CSCs are a specific subpopulation of tumor cells that possess the ability to self-renew and differentiate. In addition, these cells have been shown to express specific surface antigens and have a mesenchymal cell-associated phenotype, which is responsible for tumor differentiation, maintenance, metastasis and subsequent tumor recurrence [16]. As most cancer therapies only eliminate tumor cells, a small percentage of CSCs remain, which have the capacity to restart tumor growth due to their drug resistance and proliferative capacity [17]. Thus, the presence of CSCs in tumor tissue may explain the recurrence of many cancers. Interestingly, CSCs have been shown to express markers similar to normal stem cells, such as BMI-1 and Oct3/4 [18,19].

Recently, Colombel et al. [20] reported a subgroup of CSCs in primary PC, which were found to have a significant role in local invasion, particularly bone metastasis and seminal vesicle invasion. CSC levels have been found to impact prognosis, particularly with respect to bone metastasis risk. For example, the percentage of CSCs in PC patient bone marrow metastasis samples was shown to predict clinical outcome in bone metastases [20]. In addition, docetaxel-resistant tumor cells in PC samples are thought to be CSCs. Interestingly, a higher number of docetaxel-resistant tumor cells have been found in metastatic patient samples compared to primary patient samples. Furthermore, the percentage of CSCs in untreated primary patient samples was found to be a predictor of prognosis and biochemical recurrence [21].

Conventional therapies for PC, including ADT, radiotherapy, and chemotherapy, tend to target tumor cells with high proliferative capacity, as opposed to the androgen-independent and quiescent CSC population. Thus, following treatment, CSCs may act as an initial point of proliferation, eventually leading to tumor recurrence. Interestingly, a population of ADT-resistant cells has also been reported in PC [22]. Thus, in response to ADT, these androgen-insensitive tumor cells would survive, resulting in tumor progression. These cells, which lack or exhibit low expression of AR receptors, possess self-renewal abilities and apparent heterogeneity, and are therefore considered to be CSCs. Recent studies have focused on understanding the function of CSCs as a potential therapeutic approach to prevent tumor malignancy.

To date, only 1.5% of the human genome has been shown to encode proteins [23]. The remaining non-coding regulatory regions are defined as non-coding RNAs (ncRNAs). Moreover, ncRNAs have been shown to play an important role in various human diseases. LncRNAs, a type of ncRNA, are closely associated with the occurrence and progression of cancer [24]. Furthermore, the dysregulation of lncRNAs in malignant tumors has been closely correlated with functional changes in CSCs. Therefore, there is a critical need to examine the role of lncRNAs in CSCs to identify potential therapeutic targets for related malignancies.

This study aimed to elucidate the mechanism of MBNL1-AS1 on CSCs with a view to inhibit PC malignant progression; mRNAsi has previously been used to identify stem cell-associated tumor markers in non-small cell lung cancer (NSCLC) and validate its effects [25]. The PC stemness index mRNAsi derived by machine learning revealed significant differences between healthy and tumor tissues. Thus, mRNAsi may be a valuable tool to identify key regulators of PCSCs.

WCGNA is thought to have significant advantages over traditional differential correlation analysis methods. Here, we applied WGCNA to identify lncRNAs significantly correlated with mRNAsi among differential genes in PC and normal tissues and focused on the MBNL1-AS1 in subsequent validation and screening studies. High expression of MBNL1-AS1 has been reported to inhibit tumor progression by suppressing CSCs in colon and lung cancer [15,26]; this is consistent with the findings from our study. We used serum-free enrichment methods to isolate PSCSs from human PC cell lines. The isolated PCSCs cells were validated through the expression of stem cell markers, as well as microscopic visualization of their cellular phenotype. Next, we analyzed differentially expressed target genes in PC cells and PCSCs and found significantly lower expression of MBNL1-AS1 in tumor cells and CSCs. Furthermore, our clinical data analysis also revealed that low expression of MBNL1-AS1 had poor prognostic value in PC. In addition, it was associated with an increased risk of local invasion, lymph node metastasis, recurrence and drug resistance in PC patients. Thus, MBNL1-AS1 might be a basis for further studies. MBNL1-AS1 was found to be predominantly localized to the cytoplasm by FISH and nucleoplasmic separation assays. Cytoplasmic ncRNAs regulate expression of downstream genes by competing for miRNA binding. lncRNAs carrying multiple binding sites for the same miRNA are known as competitive endogenous RNAs (ceRNAs) [27] and participate in ceRNA networks and mRNA-miRNA-lncRNA crosstalk. These networks have been closely associated with human diseases, including cancer [28]. Our sphere-forming and western blot assays demonstrated that MBNL1-AS1 was significantly negatively correlated with the stemness of PCSCs. Subsequent in vitro and in vivo studies confirmed that MBNL1-AS1 was significantly negatively correlated with the proliferative and invasive abilities of PC cell lines and subcutaneous tumorigenesis model in nude mice. Previously, Li et al. found that MBNL1-AS1 inhibited the development of NSCLC through the regulation of miRNA-301b-3p [15], while Wei et al. demonstrated that MBNL1-AS1 inhibited proliferation and promoted apoptosis in bladder cancer cells through the miR-135a-5p/PHLPP2/ FOXO1 axis [29]. Our data suggested that MBNL1-AS1 may act as a molecular sponge, competitively adsorbing miR-221-3p to regulate CDKN1B. The miR-221-3p has previously been shown to play a critical role in the maintenance of mouse embryonic stem cells [30] and has been significantly associated with the prognosis of metastatic castration-resistant PC (mCRPC) [31]. CDKN1B, a classic oncogene that is essential for cell cycle regulation, is overexpressed in PC.

GO and KEGG analyses suggested that MBNL1-AS1 may be associated with extracellular matrix (ECM)-related pathways. The ECM is an essential component of the tumor microenvironment and is a highly dynamic structure [32]. Furthermore, the ECM can provide structural and biochemical support to promote proliferation, self-renewal and differentiation of CSCs [33,34]. GSEA analysis was used to identify the signaling pathway mediated by the MBNL1-AS1/miR-221-3p/CDKN1B axis. Our data suggested that MBNL1-AS1 and CDKN1B may both participate in the Wnt signaling pathway to maintain the normal function of CSCs. C-myc is an important target gene of the Wnt signaling pathway, and thus, upregulation of C-myc by the Wnt/β-catenin signaling pathway could restart the cell cycle process in CSCs [35,36]. Subsequent studies demonstrated that CDKN1B could negatively regulate C-myc, thereby attenuating the activity of PCSCs and inhibiting the malignant progression of PC. Prostate cancer tumor stem cells are closely related to drug resistance mechanisms. This study demonstrated that the group with low expression of MBNL1-AS1 and CDKN1B was less sensitive to chemotherapy compared to Wnt signaling pathway inhibitors, which may suggest that MBNL1-AS1 and CDKN1B can inhibit the stemness of prostate cancer tumor stem cells and thus reduce the occurrence of chemotherapy resistance.

Changes in metabolism and genotoxicity will occur during tumor progression and anti-cancer therapy, respectively. As such changes may promote tumorigenic evolution, there is an essential need to monitor changes in metabolic levels while treating tumors. CSCs have also been shown to exhibit a metabolic energy centered around the mitochondria. CSCs were found to contain fewer mitochondria and lower ROS levels, allowing them to opportunistically utilize available nutrients, as well as providing a selective advantage for their survival under extreme conditions [37,38]. Therefore, studying the metabolic mechanisms of mitochondria in CSCs has important implications for PC drug resistance. Here, by using ROS assays, we found that MBNL1-AS1 may be closely related to mitochondrial metabolism. Furthermore, WGCNA analysis identified the closely related gene SEPT4. ARTS is a pro-apoptotic protein located in the outer mitochondrial membrane [39] and is derived from alternative splicing of the SEPT4 gene, which is the only gene that can initiate the mitochondrial apoptosis process by forming this heterodimer [40].

Our study also examined the correlation between MBNL1-AS1 and the immune system. We found that MBNL1-AS1 was negatively correlated with immune responsiveness and immunotherapeutic effects. Furthermore, the tumor immune cell infiltration profile suggested that M2 macrophages were negatively associated with MBNL1-AS1 expression. M2 macrophages in PC have been shown to be significantly and positively correlated with PC prognosis [41]. Together, these findings suggested that MBNL1-AS1 may regulate the state of the immune microenvironment in PC at the immune level and improve the responsiveness to immunotherapy.

## 5. Conclusions

In summary, we found that MBNL1-AS1 could act as a molecular sponge to competitively adsorb miR-221-3p. By acting through the CDKN1B/C-myc axis, the down-regulation of MBNL1-AS1 could stimulates the Wnt signaling pathway, leading to increased proliferative and invasive abilities of PSCSs and contributing to PC malignant progression. In addition, we found that MBNL1-AS1 affected the normal energy metabolism of PCSCs and the effectiveness of immunotherapy by affecting apoptosis of the mitochondria and tumor immune microenvironment. These results suggested that MBNL1-AS1 could be a potential biomarker and therapeutic target for PC.

## Figures and Tables

**Figure 1 cancers-14-05783-f001:**
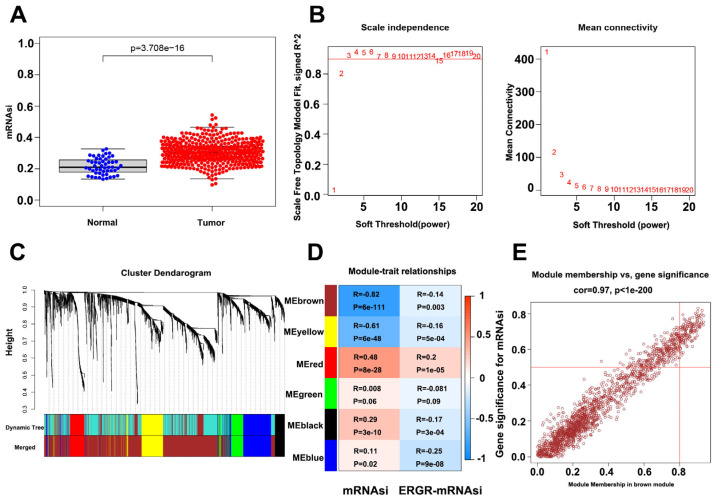
Identification of PCSC-associated genes. (**A**) Box plot showing mRNAsi scoring in tumor samples vs normal samples. (**B**) Optimal soft threshold settings. (**C**) Clustering dendrogram of 499 TCGA-PRAD samples. (**D**) Correlation index of each module with mRNAsi vs. EREG-mRNAsi. (**E**) Correlation index of genes in the brown module.

**Figure 2 cancers-14-05783-f002:**
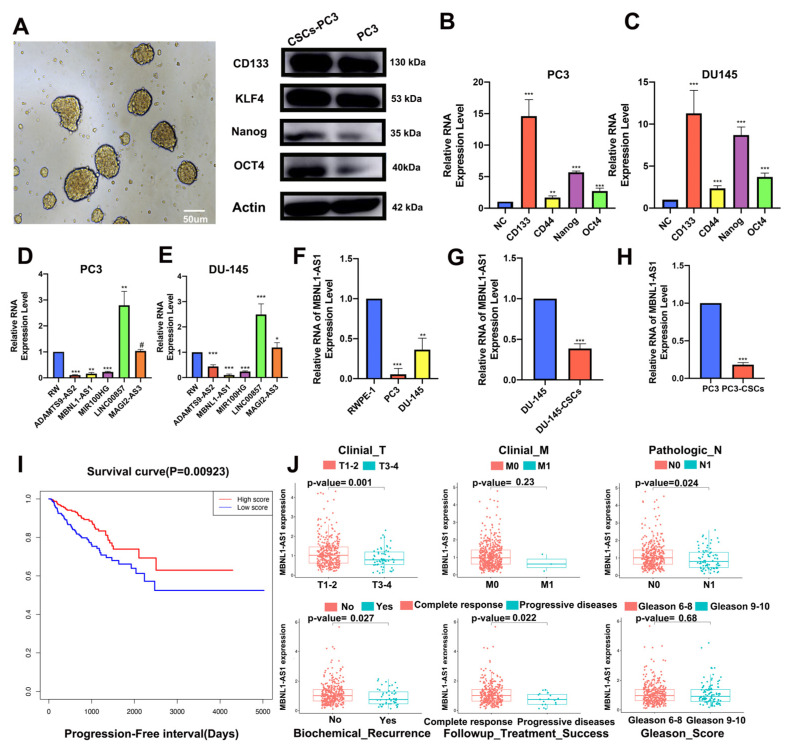
Identification of mRNAsi-related lncRNAs. PC3-CSCs were isolated and cultured under serum-free conditions. (**A**) Sphere formation, characteristic of CSCs, was visualized by microscopy. (**B**,**C**) Changes in protein (**B**) and mRNA (**C**) expression levels of stemness markers CD133, KLF4, NANOG and OCT4 were examined by western blot and RT-qPCR analyses, respectively. (**D**–**F**) mRNA expression of five candidate genes in normal PC epithelial cells and tumor cells was examined by qPCR. (**G**,**H**) mRNA expression of MBNL1-AS1 in PCSCs was examined by qPCR. (**I**) Survival analysis demonstrating the prognostic value of MBNL1-AS1 in the TGCA-PRAD dataset. (**J**) Correlation analysis showing the correlation between MBNL1-AS1 expression and clinical characteristics. * *p*-value < 0.05; ** *p*-value < 0.01; *** *p*-value < 0.001; # *p* > 0.05, respectively.

**Figure 3 cancers-14-05783-f003:**
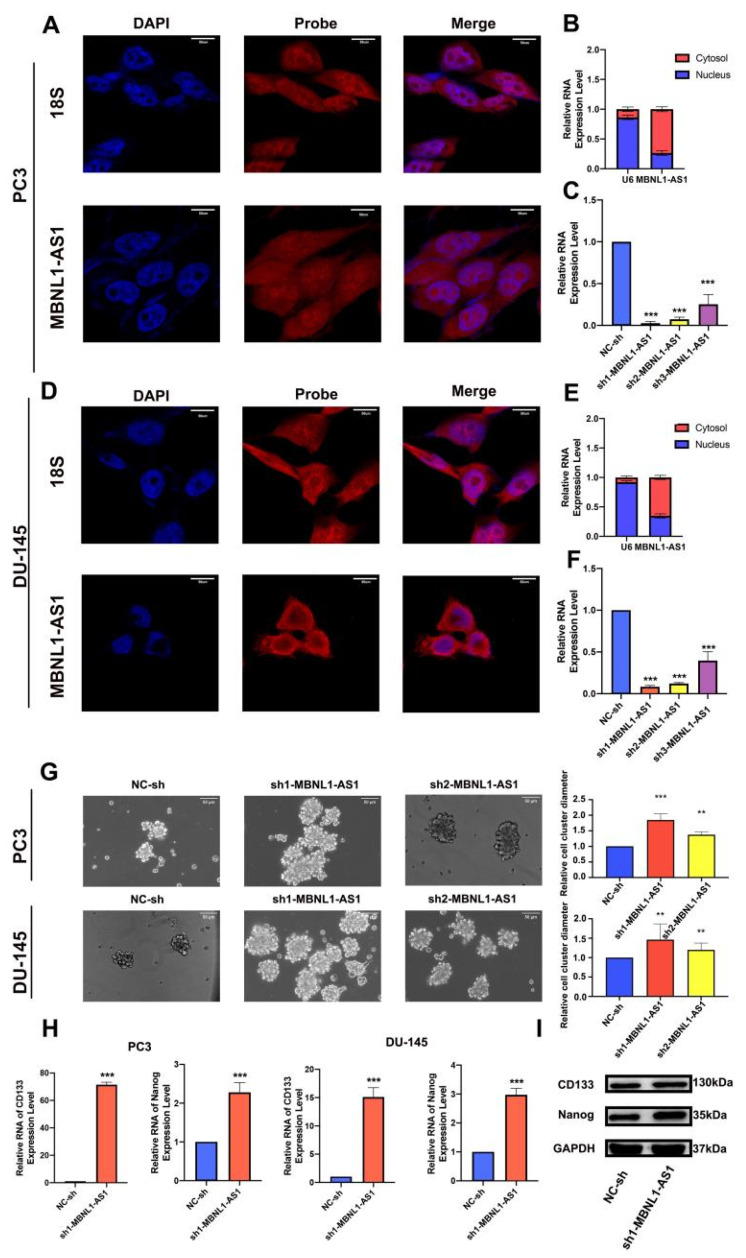
Subcellular localization of MBNL1-AS1 and its target genes, and examination of the effects of the target genes on stem cell function. (**A**–**F**) MBNL1-AS1 subcellular localization was confirmed by FISH and nucleoplasmic separation assays and sh-MBNL1-AS1 was transfected into PC3 and DU-145 cell lines, and the knockdown efficiency was determined by qPCR. (**G**) The sphere formation assay was used to examine the effect of MBNL1-AS1 knockdown on PCSC function. mRNA (**H**) and protein (**I**) expression levels of markers of stemness. ** *p*-value < 0.01; *** *p*-value < 0.001.

**Figure 4 cancers-14-05783-f004:**
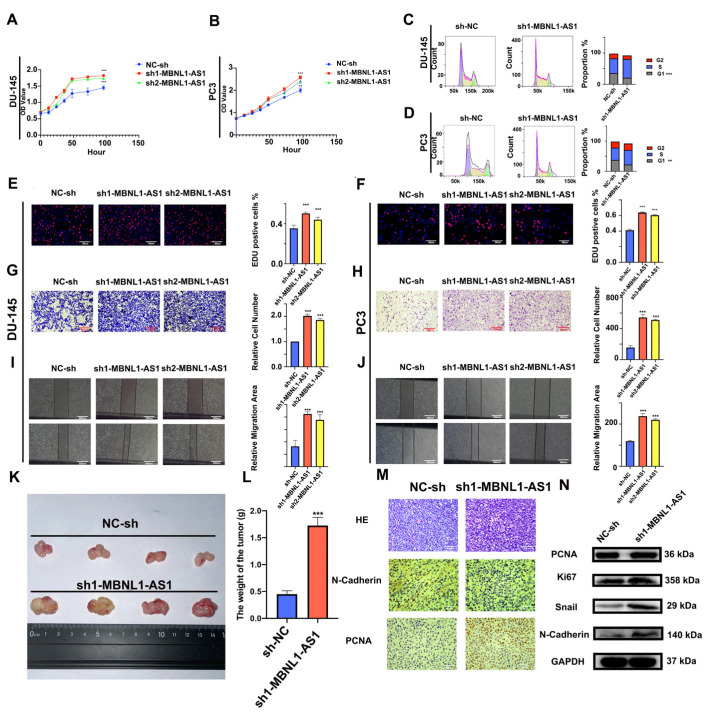
Effects of silencing MBNL1-AS1 expression on the proliferation, cell cycle distribution, and invasion of PC cells. (**A**,**B**) Cell proliferation was detected using the CCK-8 assay following MBNL1-AS1 knockdown. (**C**,**D**) Changes in cell cycle distribution were assessed by flow cytometry. (**E**,**F**) EdU assay was used to determine the effects of MBNL1-AS1 knockdown on cell proliferation. (**G**–**J**) Transwell and wounding healing assay were used to assess the effects of silencing MBNL1-AS1 on invasion. (**K**–**M**) A subcutaneous tumorigenesis assay in nude mice was used to detect changes in cell proliferation in vivo. Immunohistochemical assays were used to detect markers of proliferation and invasion. Immunohistochemical assays were used to detect markers of proliferation and invasion. (**N**) Changes in markers of proliferation and invasion were detected by western blot in the PC3 cell line. ** *p* < 0.01; *** *p* < 0.001; respectively.

**Figure 5 cancers-14-05783-f005:**
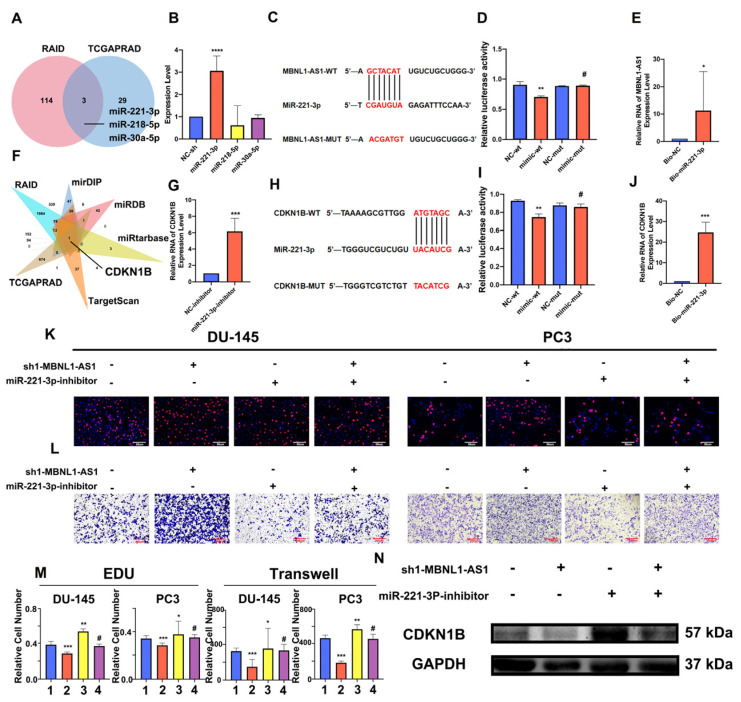
MBNL1-AS1 promotes PC malignancy through the miR-221-3p/CDKN1B axis. (**A**) The TCGA-PRAD and RAID databases were combined to predict the downstream miRNA targets of MBNL1-AS1 action. (**B**) Target miRNA expression levels were assessed by qPCR in PC3-CSCs. (**C**) The sequence of plasmid of MBNL1-AS1. (**D**,**E**) Dual luciferase (**D**) and pull-down (**E**) assays confirmed the binding relationship between MBNL1-AS1 and miR-221-3p. (**F**) TCGA-PRAD, RAID, mirDIP, miRDB, miRtarbase, and TargetScan databases were combined to predict the target genes of miR-221-3p. (**G**) miR-221-3p target gene expression levels were assessed by qPCR in PC3-CSCs. (**H**) The sequence of plasmid of MBNL1-AS1. (**I**,**J**) Dual-luciferase (**I**) and pull-down (**J**) assays confirmed the binding relationship between miR-221-3p and CDKN1B. (**K**–**M**) Rescue assays were performed to verify the effect of this ceRNA mechanism on the proliferative and invasive abilities of PC cell lines. (1. sh-NC + inhibitor-NC; 2. sh1-MBNL1-AS1 + inhibitor-NC; 3. sh-NC + miR-221-3p-inhibitor; 4. sh1-MBNL1-AS1 + miR-221-3p-inhibitor) (**N**) Changes in the expression levels of CDKN1B following decreasing of MBNL1-AS1 and miR-221-3p. * *p*-value < 0.05; ** *p*-value < 0.01; *** *p*-value < 0.001; **** *p*-value < 0.0001; # *p* > 0.05, respectively.

**Figure 6 cancers-14-05783-f006:**
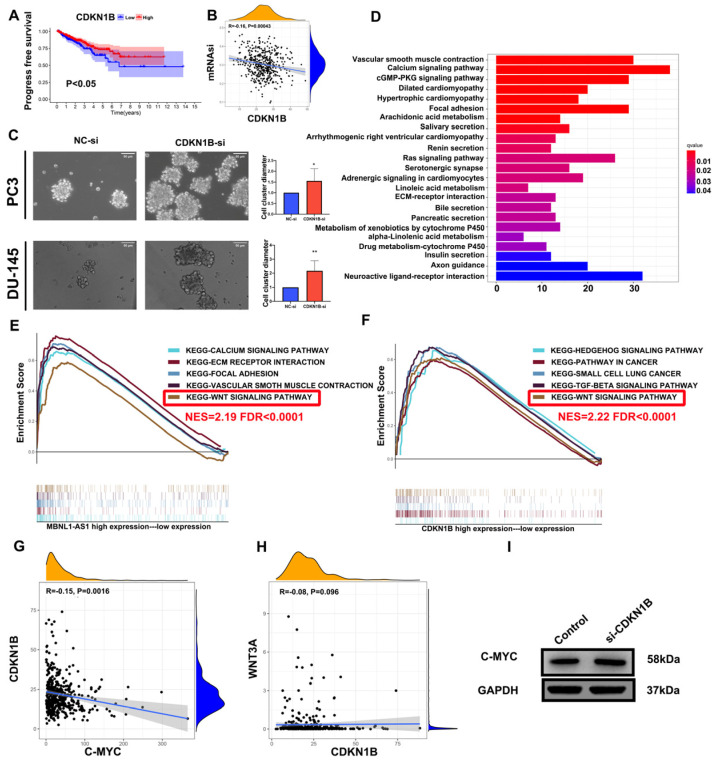
MBNL1-AS1 mediates the Wnt pathway through the miR-221-3p/CDKN1B/C-myc axis. (**A**) The KM analysis of CDKN1B. (**B**) The correlation analysis of mRNAsi and CDKN1B. (**C**) The sphere formation assay was used to examine the effect of CDKN1B knockdown on PCSC function. (**D**–**F**) The GO and KEGG analyses were used to identify the potential function of MBNL1-AS1. GSEA was used to detect the overlapping signaling pathways between MBNL1-AS1 and CDKN1B. (**G**,**H**) Correlation analysis was used to identify the relationship between CDKN1B, C-myc and WNT3A. (**I**) Western blot analysis was used to detect the corresponding protein expression levels in PC3-CSCs. * *p*-value < 0.05; ** *p*-value < 0.01.

**Figure 7 cancers-14-05783-f007:**
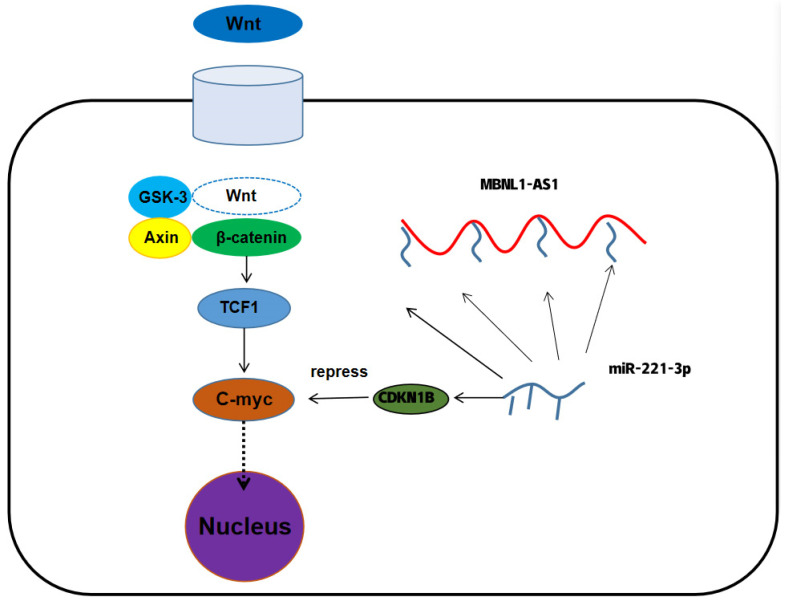
The mechanism of MBNL1-AS1 regulation of tumor stem cell function.

**Figure 8 cancers-14-05783-f008:**
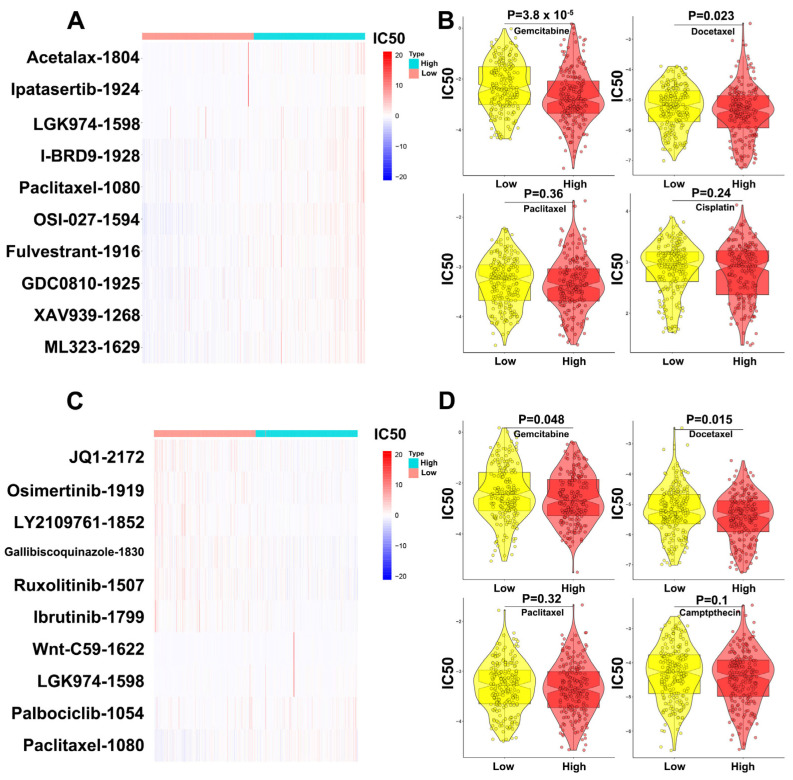
Validation of the effect of MBNL1-AS1 and CDKN1B on drug sensitivity in prostate cancer. (**A**) Prediction of MBNL1-AS1-related sensitive targeted drugs. (**B**) Prediction of MBNL1-AS1-related sensitive chemotherapeutic agents. (**C**) Prediction of CDKN1B-related sensitive targeted drugs. (**D**) Prediction of CDKN1B-related sensitive chemotherapeutic agents.

**Figure 9 cancers-14-05783-f009:**
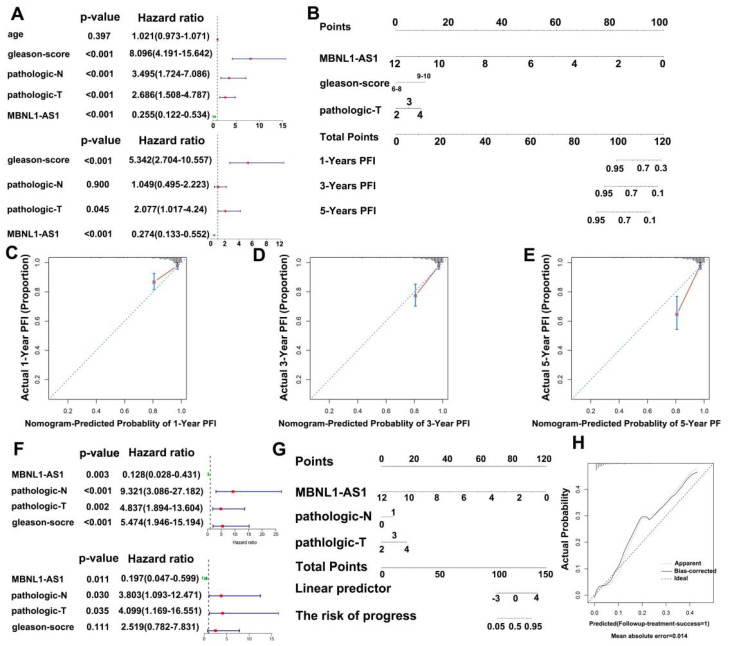
Construction of a nomogram to determine the clinical value of MBNL1-AS1 in predicting the prognosis and metastasis of PC. (**A**) Univariate and multivariate Cox regression analyses were performed to screen for variables significantly associated with PFI. (**B**) Nomogram to predict the probability of 1-, 3-, and 5-year PFI. (**C**–**E**) Calibration plots of the Nomogram were used to predict the probability of 1-, 3-, and 5-year PFI. (**F**) Univariate and multivariate Cox regression analyses were performed to screen for variables significantly associated with distant metastasis. (**G**) Nomogram to predict the probability of distant metastasis. (**H**) Calibration plots of the nomogram were used to predict the probability of distant metastasis.

## Data Availability

The data and materials in the current study are available from the corresponding author on reasonable request (drliu7087797@163.com).

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
