# Peer review of "Down-Regulation of lncRNA MBNL1-AS1 Promotes Tumor Stem Cell-like Characteristics and Prostate Cancer Progression through miR-221-3p/CDKN1B/C-myc Axis"

_cancers, 2022, doi:10.3390/cancers14235783_

Round 1

Reviewer 1 Report

Although lncRNAs have been demonstrated to be functional in different cellular activities and important in various cancers’ progression, their role in cancer stem cell (CSC) regulation still is of great interest. To identify the prostate cancer stem cells-associated lncRNA MBNL1-AS1, the authors used the single-class logistic regression machine learning algorithm, following extensive in vivo and in vitro studies to validate the findings. They were able to show that the MBNL1-AS1 inhibits prostate cancer malignancy through miR-221-3p/CDKN1B/c-MYC axis by regulating the stemness of PCSC. This study is opening a new perspective to target CSCs in order to fight the malignant progression of the prostate: cancer tumor recurrence, invasion, and drug resistance.

Overall, very important and interesting investigation, having enough material to be published in two back-to-back papers. I highly recommend accepting the paper for publication in “Cancers” with minor revisions.

 Some suggestions:

Please, increase the font size in the following Figures, as well as fix the quality of the pictures-text:  Fig.1F; Fig.2J; Fig.5A, C, F; Fig.8; Fig.9; Fig S2; Fig.S3B, Fig.S4; Fig.S4B

Some typos have been noticed:

On P.5 (Materials & Methods) the “Transwell assay” method is missing.

R.176 – ”3×105 cells.”               should be    “3×105 cells”

R.206 – “5.0×104 cells/cm2“    should be    “5.0×104 cells/cm2

R.224 – “1x104”                          should be   “1x104

R.229 –“1.0×103 cells/well”     should be    “1.0×103 cells/well”

R.250 – “8.0×106 cells”             should be    “8.0×106 cells”

Thank you!

Author Response

1、Please, increase the font size in the following Figures, as well as fix the quality of the pictures-text:  Fig.1F; Fig.2J; Fig.5A, C, F; Fig.8; Fig.9; Fig S2; Fig.S3B, Fig.S4; Fig.S4B

Reply: Thanks for your constructive comments. Thank you for your constructive suggestion, as the reviewer commented, there is real potential for further improvement in the quality of the images in the article.

Changes in the manuscript: Therefore, we have adjusted the font of the images pointed out by the reviewer in the manuscript in Fig.1F; Fig.2J; Fig.5A, C, F; Fig.8; Fig.9; Fig S2; Fig.S3B, Fig.S4; Fig.S4B.

2、On P.5 (Materials & Methods) the “Transwell assay” method is missing.

Reply: Thank you very much to the reviewers for questioning our mistake, our manuscript is indeed missing transwell-related experimental methods.

Changes in the manuscript: We have added the method of transwell assay in page 10 line 230-240.

3.Some typos have been noticed:

R.176 – ”3×105 cells.”                should be    “3×105 cells”

R.206 – “5.0×104 cells/cm2“           should be    “5.0×104 cells/cm2 “

R.224 – “1x104”                      should be   “1x104

R.229 –“1.0×103 cells/well”            should be    “1.0×103 cells/well”

R.250 – “8.0×106 cells”               should be    “8.0×106 cells”

Reply: Thanks to the reviewers for asking questions about our errors

Changes in the manuscript: We have revised these mistakes in page 6 line183; page 7 line 221; page 8 line 242; page 8 line 247; page 8 line 268.

Reviewer 2 Report

Prostate cancer is one of the most common causes of cancer death in men. In their manuscript, Liu et al. analyze the molecular mechanisms of prostate cancer recurrence and progression. They use the OCLR machine learning algorithm to analyze database samples to search for lncRNAs associated with prostate cancer stem cells (PSCS) and demonstrate the potential molecular mechanisms of PSCS-associated tumor progression. The manuscript is well structured and well written, but some corrections are needed.

Corrections:

1.       Line 191-192

Membranes are blocked with milk solution not with powder

2.       Line 214 – 222

Description of Transwell assay is wrong (copy-paste of Wound healing). Please correct it.

3.       There is no description of sphere-forming assay in materials and methods part. Please include it as separate or as a part of cell culture (growth medium, size (?)) (line 155).

4.       Line 282

Please explain at this point the choice of MBNL1-AS1 (+full name) for further analysis. This is the first time when it appears in the text

5.  Please use a larger font in the pictures

Author Response

1、Line 191-192 Membranes are blocked with milk solution not with powder

Reply: Thanks for your suggestion. We feel sorry for this inaccurate description. We will adopt your suggestions.

Changes in the manuscript: We have revised this mistake in page 6 line 200. Change milk powder to milk solution.

2、Line 214-222Description of Transwell assay is wrong (copy-paste of Wound healing). Please correct it.

Reply: Thanks for your careful checks. We are very sorry for this carelessness in methods. We will correct this mistake in manuscript.

Changes in the manuscript: We have revised this mistake in page 7 line 230-240.“Polyester membrane cell embedding dishes with 24-well plates (insert: 8.0 μm; diameter: 6.5 mm, JET, China) were used to examine the invasive and migratory abilities of PCa cells. Briefly, 6.0×104 cells were resuspended in serum-free DMEM and seeded into the upper chamber, which had been pre-coated with 200 mg/mL Matrigel (1:8, Yepsen, China). DMEM containing 10% FBS was placed in the lower chamber. After 16 h incubation, cells that had invaded the lower chamber through the Matrigel were fixed with 90% ethanol for 10 min and stained with 0.5% Crystal Violet for 15 min. Five randomly selected fields of view were observed and the number of invaded cells were counted under the microscope. This experiment was repeated three times.”

3、There is no description of sphere-forming assay in materials and methods part. Please include it as separate or as a part of cell culture (growth medium, size (?)) (line 155).

Reply: Thanks for your careful checks. We are very sorry for the carelessness of lacking a method for detecting sphere formation assay. I will add this method in the manuscript.

Changes in the manuscript: We have revised this mistake in page 7 line 213-219. “A total of 1000 cells (Including PC3 and DU145) were seeded in six-well ultra-low cluster plates (Corning, NY) and culture with DMEM/F12 serum-free medium (Invitrogen) supplemented with 20 ng/ml bFGF (Beyotime, China), 2% B27 (Thermo Fisher, USA), 20 ng/ml EGF (Beyotime, China), 5 μg/ml insulin (Beyotime, China) and 0.4% BSA (Sangon, China).A count and photograph of spheres were taken after two weeks.( Cell clusters larger than 50 mm in diameter were considered to be CSC)”

4、Line 282 Please explain at this point the choice of MBNL1-AS1 (full name) for further analysis. This is the first time when it appears in the text

Reply: The logical problem does exist, and our experiments were performed by Prostate cancer cohort, using OCLR to screen for LncRNAs closely related to CSC, and then in vivo assays to identify the target gene (MBNL1-AS1) for the study. Therefore, we adjusted the order of Figure 1A and 1B, and put these two figures into Figure S2 , thus improving the logic of the article.

Changes in the manuscript: We have revised this problem in page 9 line 297-301 and page 11 line 350-354.Furthermore we revise the Figure 1 and Figure S2 and S3.

5、Please use a larger font in the pictures

Reply: Thanks for your suggestion. We do have problems with the font size and quality of our images, and we will take your suggestions to adjust the font of the images.

Changes in the manuscript: We have revised the front of size in the manuscript in Fig.1F; Fig.2J; Fig.5A, C, F; Fig.8; Fig.9; Fig S2; Fig.S3B, Fig.S4; Fig.S4B.